# Autofluorescent Biomolecules in Diptera: From Structure to Metabolism and Behavior

**DOI:** 10.3390/molecules27144458

**Published:** 2022-07-12

**Authors:** Anna C. Croce, Francesca Scolari

**Affiliations:** 1Institute of Molecular Genetics, Italian National Research Council (CNR), Via Abbiategrasso 207, I-27100 Pavia, Italy; 2Department of Biology & Biotechnology, University of Pavia, Via Ferrata 9, I-27100 Pavia, Italy

**Keywords:** endogenous fluorophores, resilin, chitin, spectroscopy, imaging, *Drosophila melanogaster*, mosquitoes, high resolution morphology, mechanical functions, sensory perception

## Abstract

Light-based phenomena in insects have long attracted researchers’ attention. Surface color distribution patterns are commonly used for taxonomical purposes, while optically-active structures from Coleoptera cuticle or Lepidoptera wings have inspired technological applications, such as biosensors and energy accumulation devices. In Diptera, besides optically-based phenomena, biomolecules able to fluoresce can act as markers of bio-metabolic, structural and behavioral features. Resilin or chitinous compounds, with their respective blue or green-to-red autofluorescence (AF), are commonly related to biomechanical and structural properties, helpful to clarify the mechanisms underlying substrate adhesion of ectoparasites’ leg appendages, or the antennal abilities in tuning sound detection. Metarhodopsin, a red fluorescing photoproduct of rhodopsin, allows to investigate visual mechanisms, whereas NAD(P)H and flavins, commonly relatable to energy metabolism, favor the investigation of sperm vitality. Lipofuscins are AF biomarkers of aging, as well as pteridines, which, similarly to kynurenines, are also exploited in metabolic investigations. Beside the knowledge available in *Drosophila melanogaster*, a widely used model to study also human disorder and disease mechanisms, here we review optically-based studies in other dipteran species, including mosquitoes and fruit flies, discussing future perspectives for targeted studies with various practical applications, including pest and vector control.

## 1. Introduction

Biological substrates can give rise to autofluorescence (AF) emission in the near-ultraviolet (UV), visible, near-infrared (IR) spectral interval when irradiated with proper excitation light [1]. The AF emission depends on the presence of biomolecules with chemical structures suitable to interact with light, and usually comprising covalent double bonds and aromatic moieties, commonly named endogenous fluorophores. The properties of the overall light signal, in turn, will strictly depend on the chemical nature, amount, distribution and microenvironment of the various endogenous fluorophores typically present in cells and tissues under investigation, in a close relationship with their morpho-functional conditions. Autofluorescence can thus act as a valuable intrinsic biomarker useful for the set-up of real time in situ analytical and diagnostic procedures to be applied in the most various fields, from biomedicine, industry, vegetable and animal food production and processing, to environmental surveillance [2].

In all cases, the efficiency of the measuring systems for a sensitive and specific detection of the target fluorophores will rely on the choice of the optical set-up ensuring the proper excitation and emission conditions for the light-based observations. When pure compounds are available, spectroscopic and time resolved analyses can provide information on the spectral shape and time decay kinetics of their AF, to assess the most suitable optical conditions for their detection in the biological substrates where they are naturally present. On these bases, increasingly sophisticated imaging procedures have been developed, improving both excitation efficiency and specificity of the AF detection. For example, the multiphoton excitation using the sequence of photons in the red or near infrared spectral interval provides the energy equivalent to the violet or blue spectral interval, with the advantages of reaching deeper layers of the structure under study and decreasing the risk of photobleaching and UV radiation damage. The continuous development of sophisticate devices improves also the frequency and time resolution of the AF signals, advancing AF conventional imaging to multispectral, hyperspectral and lifetime imaging procedures for the detection and topological localization of specific fluorophores in living cells and tissues, up to the label-free mesoscopic applications in clinical diagnosis [3,4]. It is also worth to recall the development of procedures based on the multicomponent analysis transforming the fluorescence lifetimes in a phasor plot, which have been demonstrated to effectively solve different molecular species within single pixels. Such procedures permitted to verify the differences in the topological distribution of bound and free NAD(P)H and of FAD in mammalian cell models with a different engagement in aerobic or anaerobic energy metabolism [5].

In insects, optical phenomena have deserved great attention since a long time. This is the case of Lepidoptera and early investigations on their wing fluorescing pigments, commonly referred as papilliochromes, and their components, identified with the support of combined optical analyses of compounds purified from natural extracts and autoradiographic imaging and chromatographic assays following administration of ^14^C labelled dopamine and tryptophan [6]. Numerous additional studies focused on the chemical identification of the pigments responsible for AF emission, as well as on effects like reflection and diffraction induced by the grating-like structures of the ribs located along the ridges of the scales, acting as photonic crystals (Figure 1) [7,8,9].

The characterization of AF and color properties has prevalent implications as a support in taxonomical classification [11,12,13,14,15], while the studies on the mechanisms underlying the interaction of photonic structures with light and their responses to external factors have inspired the set-up of sensors for applications ranging from energy accumulation to the monitoring of environment and human health [16].

Similarly, in Coleoptera, the porous layers structurally organized in ordinated and periodic stacks localized in the cuticle of thorax and elytra have been regarded as photonic cells, and their ability to sense the refractive index of fluids by means of changes in light reflection or fluorescence has inspired the set-up of fluid biosensors [17,18,19]. In this respect, it is worth also to recall that reflectance spectra measured on adults of two beetle species (*i.e*., *Sitophilus zeamais* Motschulsky and *Cynaeus angustus* (LeConte)) with sclerotized exoskeleton allowed to monitor their response to treatment with killing agents, demonstrating the potential of the technique for further applications as a non-destructive and non-invasive approach to assess stress conditions in insects [20].

With specific reference to the biomolecules responsible for AF emission, besides the abovementioned fluorescing pigments of Lepidoptera, resilin and chitinous compounds have been greatly investigated with attention to their structural and functional implications. Resilin is the very resilient elastomeric, or rubber-like, protein detectable in various arthropod body structures, where it ensures important biomechanical properties such as elasticity and energy storage in the pterothorax cap tendons in the direct flying apparatus of Odonata [21,22,23,24]. The typical bluish AF emission of resilin favors the detection of structures with prevalently elastic properties, helping to distinguish them from material with different degrees of sclerotization. Chitin as commercial compound purified from natural sources has been demonstrated both to be practically non-fluorescent [25] or to give rise to an emission in the 400–550 nm range [26]. The absence of AF from pure chitin is consistent with its polysaccaride nature. However, in a chitinous material, a variable presence of oxidised and undefined compounds can be responsible for the AF emission and the lenghtening of the wavelenght position. This suggestion is supported by an AF lifetime study performed on the cuticle of *Cimex lectularius* Linnaeus (Hemiptera: Cimicidae) [27] that reported the absence of long decay times relatable to chitin [26] On the other hand, the detection of decay times of 0.4 ns and 1.0–1.5 ns indicated the presence of resilin along with other fluorophores such as melanin. On this basis, Reinhardt and colleagues proposed the possibility to distinguish areas rich or poor in resilin, respectively [27]. The progressive variation in the reciprocal content between chitinous components and resilin has been proposed to be directly reflected by the changes of AF emission from blue, yellow and green to reddish colors, detected by means of confocal microscopy at properly selected excitation/emission wavelenghts [22]. Comparable AF measuring conditions have been also used by Eshghi and colleagues for the development of an algorithm based on a modulus color map able to assign each color of a confocal AF image to a specific type of material [28]. In this way, it is intended to improve and facilitate the investigation of the relationships between the biochemical components and functions of the body structure under investigation, as shown by the results obtained on different insect species. This is the case of *Coccinella septempunctata* Linnaeus (Coleoptera: Coccinellidae) and the tarsal seta, *Cassida rubiginosa* Muller (Coleoptera: Chrysomelidae) and the male flagellum, *Carausius morosus* (Sinéty) (Phasmatodea: Lonchodidae) and its hindleg tibia [28]. In addition, in the leaf beetle *Chrysomela populi* Linnaeus (Coleoptera: Chrysomelidae), the combined morphological and AF-based analysis allowed the identification of two subtypes of gustatory sensilla chaetica on the antennal apical flagellomeres. The structures were classified in two types, those fluorescing prevalently in blue, ascribed to the dominance of resilin ensuring flexibility, and those fluorescing prevalently in green, attributed to the stiffer chitin. The latter type, the subtype 2, has been proposed to sense both primary and secondary metabolites of the host plant by contact chemoreception, with promising insights for in situ investigations on the capacity of phytophagous insects to sense plant biochemical components [29].

Last, but not least, it is worth mentioning that AF rising from various tissues and organs of insects can often disturb the identification of target sites labelled with specific exogenous fluorescing dyes or the expression of fluorescent proteins introduced with genome manipulations. This issue is similar to what frequently reported in the case of fluorescence-based histochemical assays in biomedicine, with particular relevance when the targets consist in few and scattered positively labelled sites [2].

Technical procedures have been thus developed to treat the specimens in order to minimize the AF signal. In some cases, a proper decrease of AF can help the topological localization of the investigated target sites. This possibility has been elegantly exploited by Pende and colleagues, who developed a combined tissues clearing procedure to decrease the AF with an ultramicroscopy approach (Figure 2). The resulting lowered signal of AF allowed to delineate the body of *Drosophila melanogaster* Meigen (Diptera: Drosophilidae) larvae and adults, favoring the multi-view topological localization of the green fluorescent protein (GFP)-expressing neuronal network with single-cell resolution in *Drosophila* larvae [30].

In insects, AF was also reported as a real disturbance through confounding and impeding the reliable detection of the fluorescent signal from the reporter gene [31], or limiting the detection of 16S rRNA-targeted fluorescent in situ hybridization (FISH) applied to tissues for the analysis of symbiotic bacteria [32,33]. The disturbance can be due mainly to the presence of food particles in the gut, or to Malpighian tubules, chitinous exoskeleton and necrotic tissues, responsible for high AF signal that could be mistaken for the fluorescence of the introduced markers, reducing or even hampering their specific detection [34,35]. In this context, efforts have been devoted to circumvent the problem. Different optical clearing methods and microscope imaging procedures have been compared, in order to improve the localization of *Plasmodium* parasites in the midgut of *Anopheles* vectors [36]. On the other hand, Koga and colleagues have developed a protocol based on the use of hydrogen peroxide able to strongly reduce AF in insect tissues and allow a reliable FISH detection of endosymbionts of aphids, lice and bat flies [37]. Anyway, several studies performed in insects on the development of transgenic individuals through germline transformation and consequent expression of fluorescent protein markers (e.g., green and red fluorescent proteins) have reported that AF signals did not impair the detection of the experimentally-induced fluorescence [28,29,30,31,32,33,34,35,36,37,38,39,40,41,42,43]. It is also worth to recall the particular case of the AF rising from the yolk granules and the vitelline membrane of dechorionated *D. melanogaster* embryos, which, together with the embryo thickness, size, and opacity, seemed to affect the application of optical fluorescence microscopy in the study of this developmental stage. However, the lower AF signal obtained from yolk and vitelline membrane in the green excitation condition (543 nm) than in the blue one (458 nm, 488 nm; two photons 820 nm), along with the AF localization restricted to specific structures, was exploited to exclude the AF hindering the detection of exogenous dyes or to provide an optical guide to detected specifically labelled sites [44]. In the same paper, additional strategies to avoid AF-related issues consist in the use of optical sectioning by confocal laser scanning microscope and by photobleaching or photoactivation procedures.

Autofluorescence in Diptera has been investigated for variegated purposes and in a much more scattered manner than in the abovementioned insect orders of Lepidoptera and Coleoptera. Therefore, this review aims to provide a comprehensive and up-to-date overview about the diverse AF-based studies in Dipteran species, with particular attention to their most relevant endogenous fluorophores, not excluding other optically-based phenomena. The content of this review is summarized in Figure 3.

## 2. Autofluorescence in Diptera

An early attempt to investigate AF under UV light excitation was promoted in the genus *Phlebotomus* (Diptera: Psychodidae). With the aim to circumvent the use of radioisotopes and related economical and management issues, as well as insect vitality loads, the observation of various AF patterns in different organs was successfully reported [45]. A next work on the AF of Diptera proposed confocal microscopy as a relatively simple and reliable means to assess and describe their structures, with the advantage to evidence details otherwise hard to distinguish by bright field imaging. Acquisition of AF images under the common excitation at 543 nm permitted the appreciation of the fine structures of the head, in particular mouthparts and antennae, and genitalia in three dipteran families, namely Campichoetidae, Camillidae, and Drosophilidae [46]. Remarkably, this work promoted the use of AF imaging applications as a support to facilitate structure identification and sharing of morphological, phylogenetic and taxonomical data among entomologists. Even more importantly, the study predicted also the helpful role of AF analysis for the interpretation of the functional role of soft structures as compared with the most sclerotized ones. This prediction has been gradually taken shape even exceeding the promised expectations, as illustrated by the following reports testifying the increasingly deserved attention to the fluorescing structures of Diptera, with consequent valuable implications for the study of mechanical and sensory functions, such as olfaction and hearing, metabolism in different tissues and developmental stages, including the response to different physiological states.

### 2.1. Mechanical Functions

Autofluorescence imaging analysis has greatly helped the characterization of the attachment system of the bee lousefly *Braula coeca* Nitzsch (Diptera: Braulidae), a kleptoparasite on the honeybee *Apis mellifera* Linnaeus (Hymenoptera: Apidae). The attachment system to the bee’s hairy surface consists in tarsal appendages, with a pair of claws and two pulvilli. Images obtained by confocal microscopy at selected excitation/emission conditions showed AF patterns with differently colored areas (Figure 4). The blue regions indicated a dominating presence of resilin in the pulvilli, consistently with their soft mechanical properties ensuring adhesion to soft surfaces, while the yellowish, greenish and reddish regions were put in relation with increasing degrees of sclerotization, and thus of stiffness, in the tarsus and its structures, such as claws and grooming setae [47].

These findings are in a remarkable agreement with a previous report on the attachment system of the leg of *Crataerina pallida* (Latreille) (Diptera: Hippoboscidae), a non-flying ectoparasite of birds, in particular of the swift *Apus apus* (Linnaeus). In this study, the authors showed the presence of a material gradient with prevalence of the sclerotized chitinous-like components in the basal region of the claws, the pulvilli and the empodium, and a gradual change to the soft resilin apical tips of the setae, ensuring the functionality of the attachment system to allow adhesion of the ectoparasite to the bird feathers [48].

In adults of *D. melanogaster*, resilin has been shown to contribute as a protein matrix to skeletal structures, organs and tissues that are involved in repeated movements, torsion or flexion [49]. In particular, resilin-enriched sites include spots at the leg joints, at the articulation of the wings (where resilin patches may act as muscle attachment sites) and on the abdominal flanks, in cibarium and labellum in the head, in the tracheal endings, at hair bases and in the spermathecal ducts, as observed in mosquitoes [50,51,52,53]. A recent study focusing on the analysis of resilin in the spotted-wing fruit fly *D. suzukii* (Matsumura) (Diptera: Drosophilidae) showed that the intensity and distribution of resilin signals are conserved with respect to *D. melanogaster* [54]. However, in the wing hinge and in the trochanter, the resilin signal is stronger in *D. suzukii* than in *D. melanogaster*, while there are no differences between *D. suzukii* and *D. hydei* Sturtevant (Diptera: Drosophilidae). These findings are indicative of potentially different biomechanical features responding to different lifestyles, and further investigations will be essential to clarify the functional adaptation of these structures.

### 2.2. Sensory Functions

The interpretation of AF-based data in the characterization of the material composing specific structures, besides being relevant to study mechanical functions as mentioned above, is increasingly deserving attention to describe sensory processes, which are target of intensive research across Diptera [55,56,57,58,59,60,61,62,63,64]. Sensory processes take place at the interface between insects and their environment. Therefore, the understanding of their molecular and physiological bases, together with the biomechanical features of organs and tissues, will allow gathering information essential for both basic and applied research. Sensory processes comprise, but are not limited to, chemoreception (including olfaction and taste), hearing and vision.

#### 2.2.1. Chemoreception

Autofluorescence observation allowed to approach the description of the sensilla in the antennae of adult individuals of the Asian tiger mosquito, *Aedes albopictus* (Skuse) (Diptera: Culicidae) [65]. Along with the flagellomers in the female antenna, AF was detected in sensilla trichoidea, the primary olfactory sensilla, and in the short-grooved peg sensilla. In the male antennae, AF is clearly visible at the terminal segment, both at the terminus, where the campaniform organs are known to be localized, and in other sensilla, which require further investigation. Moreover, a recent study combining imaging and spectrofluorometry described, for the first time, AF signals in the larval head of two mosquito species of key public health importance, *Ae. albopictus* and *Culex pipiens* Linnaeus (Diptera: Culicidae) [66]. The AF in both mouth brushes and antennae, while being generally conserved in terms of distribution patterns, revealed interesting differences potentially related to the biology and ecology of the two species. For example, the blue AF attributable to resilin at the antennal bases is more extended in *Cx. pipiens* than in *Ae. albopictus*, maybe to reflect different engagement in antennal movements. The AF spectra also suggest different material properties that it is essential to further investigate from the functional point of view.

As far as it concerns gustatory function, it is interesting to note that in the *Drosophila* wing anterior margin gustatory neurons’ dendrites are located inside long and thin bristles, and an elegant strategy has been developed to study the wing gustatory hairs in this species [67]. Since the tracheal tubes localized in the wings are filled with air, the ability to collect and drive internally volatile substances has been investigated by using vapors of benzaldehyde, taking advantage of induced fluorescence produced when this fixative binds covalently to proteins. In fact, the exposure to the volatile benzaldehyde induced a marked increase in the fluorescence emission of the chemosensory hairs localized at the anterior margin area of the wings, demonstrating a capillary-like condensation mechanism allowing the volatile compound to be transferred in the hair (Figure 5).

This work has thus provided a direct demonstration that the air turbulence produced by the wings, along with the structures of capillaries favoring their fluid dynamics, can account for the chemosensory functions of hairs and tracheal pipes. In this respect, a reciprocal benefit has been proposed between the use of innovative nanotube technology, expected to improve the understanding of insect sensory physiology, and the deeper characterization of insect sensory machinery to inspire new technological solutions.

#### 2.2.2. Hearing

In insects, hearing organs can be generally divided in tympanal ears sensitive to the sound pressure component and movement receivers responding to air particle oscillations in the sound field [68]. Organs involved in the role of movement receivers are commonly light structures such as antennae and sensory hairs, able to be deflected by the oscillations of the air particle pressure produced by sounds.

Recent studies are starting to investigate the material composition of the antennae in adults of different dipteran species, especially mosquitoes and midges, to achieve a deeper understanding of acoustic sensing. These studies have taken advantage of AF-based imaging and spectrofluorometry, as well as of approaches uncovering antennal mechanical behavior. The prevailing blue AF ascribable to resilin in relatively soft structures has been explained according to the role of this protein in ensuring flexibility and protection against the mechanical stress antennae are subjected to. A comparative analysis between male and female of the midge *Chironomus riparius* Meigen (Diptera: Chironomidae) [69], and a study between the mosquito species *Toxorhynchites brevipalpis* Theobald and *Anopheles arabiensis* Patton (Diptera: Culicidae) [70] have demonstrated a sequence of bands containing soft and hard chitinous material along the antennae, according to the blue and yellow-green AF regions detected by confocal microscopy. With the support of a computer simulation, this banded structural organization has been indicated to ensure the ability to tune the response of the antenna to sound vibrations, beyond what could be expected considering only the mere shape of a model structure composed by homogenous material. These findings contribute to improve the knowledge on the evolution, ecology and behavior of dipteran species, likely helping the development of strategies for the control of pests and vectors, besides inspiring the development of vibration sensors.

A study has followed, based on both AF imaging and spectrofluorometry, aimed at characterizing the fluorescing structures in the head appendages and body scales of adult males and females of *Ae. albopictus*, a vector species up to now poorly investigated from the optical perspective. The AF patterns have shown differences suitable to improve the knowledge on the sexually dimorphic body compartments of the species [65]. The antennae, in particular, have shown that the thirteen flagellomers present in both sexes exhibit a different structural organization as to both the flagellomer joints and the distribution of fibrillae and sensilla. Remarkably, the female antennae carry sensilla along their entire length, and AF rises from each antennal joint. Here AF allows to appreciate both a thin disc and a larger cone, which in turn has been demonstrated by spectral analysis to fluoresce at a slighter longer wavelength as compared to the bluish emission of the thin disc. In the male, on the contrary, the segments of the 1–11 flagellomers carry long fibrillae and exhibit a strong bluish emission, along with a very thin blue disc, and sensilla can be observed only in the two terminal segments. An example of the AF potential to improve the appreciation of sexual dimorphism in the flagellomer joints, in terms of material composition besides structural differences, is shown in Figure 6.

In the light of the previous reports by Saltin and colleagues addressing the importance of antennal components in influencing its mechanical response to sound vibrations [69,70], the findings on the AF-based antennal sexual dimorphism in *Ae. albopictus* provide further support for new studies on intra- and inter-specific communication abilities.

#### 2.2.3. Vision

Insect vision has deserved great attention since decades, as proven by an extensive literature comprising reports ranging from eye structural organization, neuronal mechanisms in color vision, visual receptors, photoactive and antioxidant biomolecules, to the behavioral implications entailing feeding and mating [60,71,72,73,74,75,76,77].

Photophysical properties of photoreceptors in insects, in turn, have been widely investigated as to both AF emission, light filtering, visual pigments and spectral responses [78,79,80,81,82,83,84,85,86]. Of note, *Drosophila* is the preferred model of the majority of these studies dealing with photoreceptor mechanisms and visual pigments, taking advantage of the solid knowledge of its biology, genetics, genomics, a standardized rearing, the availability of wild-type flies and well characterized mutants [87].

Given the wide literature covering the description of the various aspects of vision in insects, according to the abovementioned reports, here we discuss only few selected aspects on AF and color sensing with related biomolecular mechanisms and behavioral issues, which have been greatly investigated in *Drosophila* and mosquitoes.

The high efficiency of rhodopsin in using light for photoisomerization as compared with the activation of other photophysical effects, including AF emission, is in agreement with the economy of optimizing the use of radiation energy to initiate the vision process. Autofluorescence, indeed, is commonly reported to be produced as a red signal from metarhodopsin, which is the molecular species resulting as a photoproduct of the light-induced conversion of rhodopsin in the sequence of changes from the photoreceptor membrane potential to the eventual generation of nervous stimuli [79,88,89]. The red AF characterizing metarhodopsin, as compared with the poorly fluorescent rhodopsin [90], allows its direct *in situ* detection, useful for consequent valuable applications. Fluoroscopy methods developed to investigate the eye photoreceptors have been successfully used in studies on the mutation effects on *Drosophila* ommatidia differentiation, taking advantage of the AF of the photoreceptors, and in particular of the red fluorescing metarhodopsin (Figure 7) [91].

The analysis of metarhodopsin AF, combined with electroretinograms, also enabled to study the rhodopsin-arrestin cycle in the visual response to light wavelength and intensity in *Drosophila*. Based on the comparison of wild and mutant individuals, namely the arrestin2-mutant deficient in arrestin, which is the inactivating agent of metarhodopsin, Belušič and colleagues have indicated that the transition between repolarization and depolarizing afterpotential may effectively depend on both the intensity of light and arrestin concentration [93]. These findings provide a tool for the estimation of the ratio between arrestin and rhodopsin under different experimental conditions and subsequent implications for the investigation on the vision process. The red AF rising from photoreceptors has been also exploited as a valuable intrinsic spatial optical biomarker to validate the application of a micro-ophthalmoscope for the in vivo, noninvasive visualization of the retina, assessment of refractive state and optical organization and focusing ability in spiders and flies [94].

Promising perspectives are provided by the studies on the influence of insect color visual sensitivity on their behavior, with implications for the control of vector species. For example, the sensing of CO_2_ has been demonstrated to strongly favor the attraction of mosquitoes for the orange and red color bands, matching with the human skin optical properties. In fact, blocking the ability to detect CO_2_ or color filtering has proven to prevent the targeting of the skin, indicating the strict interaction between olfaction and vision in mosquito feeding behavior. This finding has thus been proposed as a promising basis for the set-up of strategies to prevent mosquito attraction to human hosts [95].

Similarly, the use of LED light sources emitting at different UV wavelengths has been investigated as to the ability to attract nocturnal mosquitoes, with the aim to improve trapping of these insects for both surveillance and vector control purposes [96].

Regarding the control of agricultural pests, promising insights have been provided by a recent study on the description of both the ultrastructure and the optical properties of white patches present on the head and thorax of adults of the olive fruit fly *Bactrocera oleae* (Rossi) (Diptera: Tephritidae). A UV-induced blue AF, potentially due to resilin content, has been detected in white patches, suggested to derive from modified tracheal air sacs below a transparent cuticle (Figure 8). 

These air sacs are proposed to constitute a 3D photonic solid structure or crystal, able to scatter and reflect light contributing to the effective white aspect, likely playing a role as visual cues in sex recognition and/or predatory avoidance. The possible involvement in sexual recognition has been thus suggested by the sexual differences in the reflectance spectra of the scutellum, further pointing at the need to better explore the role of visual cues in lekking and mating behavior in these and other tephritid species, also with perspectives for the set-up of pest control strategies [97].

Visual signals in Diptera rely not only on color patterns, depending for example on the topological distribution of the mosquito body scales in which white, black, and reddish colors can be perceived, along with different fluorescing properties [65], but also on elaborate structural shapes, including patterned wings that are used to generate a wide set of movements. In Tephritidae, the surface of the wings is turned towards conspecifics in a complex and specialized communication vocabulary. For example, the sexually dimorphic face (i.e., prefrons), which is white in males of *Ceratitis capitata* (Wiedemann) (Diptera: Tephritidae), has been regarded as an important player in visual communication [98]. The surfaces, i.e., the prefrons and the wings, of *C. capitata* and *Anastrepha suspensa* (Loew) (Diptera: Tephritidae) have shown sexual differences in their UV reflectance and translucency properties, enhanced when they are observed against a highly reflective background, opening the avenue for further analyses of this phenomenon in insects [99].

These findings, in view of the above-described studies on the biomechanical properties of antennae influencing hearing functions, can been proposed as a novel perspective to further clarify communication strategies in these pests and disease vectors, valuable for the set-up of new strategies for in-flight detection and tracing of detrimental insect species.

### 2.3. Metabolism

The AF-based metabolic studies in insects have been greatly based on the detection of NAD(P)H and flavins. These coenzymes are indispensable for the oxidoreductive reactions. Their ability to fluoresce in the respective reduced and oxidized state has been since a long time exploited to investigate the functionality of the anaerobic and aerobic energy pathways of energy production, as well as for the reductive biosynthesis and defense against oxidizing species [100]. Besides the spectral shape properties, the shorter AF lifetime of free NAD(P)H, as compared with that of the bound one, has been proposed as a useful parameter to investigate the cell engagement in anaerobic and aerobic metabolism [101].

In general, the selective measurement of AF at different intervals during the decay of the emission signal following the excitation pulse is a valuable tool for the simultaneous detection of different fluorophores and lifetime imaging technique (FLIM) in cells, tissues and organs for analytical and diagnostic purposes [4]. In fact, many studies aimed at characterizing the morpho-functional properties of the most variegated biological substrates take advantage of the possibility to separate different endogenous fluorophores/exogenous dye markers/fluorescent proteins expressed in genetically manipulated cells and animals depending on the lifetime of their emission signals. This possibility, associated with the advances in luminescence imaging, contributes to provide 3D spatial resolution [3].

In Diptera, and in particular in the *Drosophila* model, the optical assay of the intrinsic AF of NAD(P)H and FAD by means of a two-photon FLIM was applied to characterize the metabolic states of different tissues, including larval salivary glands (Figure 9A,B,F,G) and fat body cells (Figure 9C,D,H,I) and adult female midgut enterocytes I (Figure 9E,J), to validate the procedure for the investigation of the sperm metabolism [102].

Sperm has been characterized for the first time *in situ* in its natural environment. Interestingly, Wetzker and Reinhardt found that *Drosophila* sperm had a predominant glycolytic energy metabolism [102], similarly to what reported for human and rodent sperm [103,104]. This finding is also in agreement with an early report on the limited presence of enzymes of the mitochondrial respiratory chain in *Drosophila* stored sperm, as demonstrated by ultrastructural cytochemical assays [105]. In addition, while no differences were reported between male- and female-stored sperm when tracing NAD(P)H lifetimes, the fraction of protein-bound FAD was significantly lower in female- than in male-stored sperm, demonstrating a metabolic plasticity in adapting to the environment, resulting in a higher metabolic activity in sperm located in the females [102]. These data support the idea that sperm undergo metabolic alteration(s) after transfer to the female seminal receptacle, a specialized organ where sperm can be stored and released for fertilization [106], and open valuable paths to investigate sperm functionality. The possibility to assay sperm metabolism in a label-free manner, *in vivo* and directly in its physiological environment, is particularly relevant for comparative studies on sperm metabolism across insect species, as shown in Hemiptera [107,108], but also to set-up protocols to apply to the seminal fluid of higher animals and humans. In these regards, FLIM was recently used to assess the rate of oxidative phosphorylation in *Drosophila* sperm [109]. In this study, aimed at evaluating the effect of male aging, the authors detected higher metabolic rate and increased mitochondrial H_2_O_2_ levels in the sperm from older males and in comparison to the gut epithelial tissues. The finding that sperm from older males have a more rapid metabolism and more mitochondrial ROS is promising in the field of fertility, also with respect to human infertility treatments. Similar technical approaches are showing to be effective in targeting not only adults but also embryos and larvae of dipteran species to address both structural and metabolic questions (see [44] for a review). In the *Drosophila* embryo, the use of a phase shaped laser pulses, and tuning of two photon excitation at the blue and at the red side of the 700–950 nm range allowed to separately detect the enhanced green fluorescent protein (eGFP) and the endogenous fluorescence, this latter ascribed to NADH. The proper tuning of the excitation wavelength and selection of emission has been thus proposed as a valuable strategy to improve a multiplex fluorescence microscopy by using a single laser [110]. A particular use of AF consisted in its exploitation as a reference emission signal to guide the measurement and correction for the optical aberration in thick structures with dishomogeneous components. In this view, this strategy applied to the *Drosophila* embryo has demonstrated the possibility to collect reliable live images from the middle of the embryo even at its early developmental stages, allowing to trace mitotic processes [111].

Autofluorescence detected upon two-photon excitation has also supported a metabolic investigation in the fat body at the pupal stage during metamorphosis in *Drosophila*. The remodeling of the fat body, consisting in the dissociation into distinct fat cells, has been monitored by means of the anti-Stokes Raman scattering-based technique. This latter, able to image directly the lipids in the fat body of larvae and pupae, has shown a decrease in lipids, likely metabolized to provide energy for metamorphosis [112]. On the other hand, an increase in AF has been ascribed to kynurenine rather than to the fluorescing pteridines, with the support of chromatographic analysis [113]. The more marked AF phenomenon, involving also a larger number of cells observed in larvae fed with an excess of tryptophan, has indicated its relationship with the increase in adaptative, inducible enzymes, with particular reference to the tryptophan pyrollase system [112,114,115].

Imaging of the AF in granules in the Malpighian tubules of *Drosophila* mutant models, detected by confocal microscopy under 405 nm excitation in combination with the emission of the GFP and DAPI counterstaining of the nuclei, has supported also the role of 3-hydroxykynurenine, a tryptophan metabolite, in the storage and regulation of zinc metabolism [116]. Since zinc transporters are evolutionary conserved, this model has thus been proposed as a valid tool to improve the understanding of mechanisms underlying zinc metabolism in higher animals and humans. The particular relevance of this finding is also relatable to the role of zinc as protein and enzyme cofactor in the regulation of important functions, including immunity, and the negative outcomes of zinc deficiency, such as its association with low protein intake.

Metabolic pathways underlying the maintenance of the functional components and structures across cell life also result in the production of waste and oxidized compounds. These compounds can derive from the digestion of organelles not released by the cells, and can contain, in variable proportion, oxidized lipids, proteins, glycoproteins, carotenoid and porphyrin derivatives, with different degrees of oxidation, crosslinking and aggregation. These products are commonly defined as lipofuscins and consist in intracytoplasmic fluorescing granules that accumulate during aging in the cytoplasm of cells of mammals and Arthropoda, including insects [117,118]. The heterogenous composition of lipofuscins accounts for the variability of their fluorescence spectral emission properties [119,120,121,122]. In Diptera, the early studies by Sheldahl and Tappel [123] demonstrated the presence of fluorescing compounds in aged *D. melanogaster*, ascribed to peroxidase lipids. Similarly, Miquel and colleagues found that aged *Drosophila* tissues contain fluorescing lipopigments, with maximal excitation at 370–375 nm and emission detectable in the 440–450 range, similarly to the AF of lipofuscins reported from aged mice [124]. The age-related increase in lipofuscins was also shown in male adults of *Musca domestica* Linnaeus (Diptera: Muscidae). Of note, in the house fly, a greater number of orange fluorescing granules was detected in the oenocytes than in fat cells of the midgut epithelium, and in the head and thorax compartments, indicating the presence of oxidized polyunsaturated lipids able to react with proteins and nucleic acids to result in Schiff-base fluorescing products [125,126]. These reports surely contributed to the understanding of the biochemical origin of lipofuscins, with promising perspectives for additional investigations. However, to the extent of our knowledge and despite their potential as markers of aging and oxidative events, lipofuscins have up to now deserved poor consideration in insects [127]. A renewed interest in lipofuscin AF detection is highly desirable to improve aging or stress response studies on Diptera.

Pteridines are a group of products of the purine metabolism, for which a primary role as cofactors in hydroxylation reactions has been proposed as an initial physiological reaction common to almost all organisms, followed by various pathways [128]. In insects, the main biological function ascribed to pteridines consists in the inactivation and excretion of nitrogen-containing catabolites, or in the synthesis of pigments contributing to the livrea, with signaling and behavioral implications, and as components of insect eye [128,129,130]. The AF of pteridines has been exploited for their direct localization in the fat body of third instar *Drosophila* larvae, in the cells of the posterior region, contrarily to kynurenine, whose AF is detected in the cells of anterior region [115]. This finding has suggested the physiological ability of the cell to differentially produce pteridine or kynurenine, supported also by the abovementioned ability of the so-called pteridine-cells to begin to synthetize kynurenine following larval tryptophan overfeeding [115]. Pteridines are known to accumulate in insect eyes during aging [128]. This trend has been shown in multiples dipteran species, including *D. serrata* Malloch (Diptera: Drosophilidae) [131], *Calliphora vicina* Robineau-Desvoidy, *Chrysomya bezziana* Villeneuve, *C. megacephala* (Fabricius)*, Cochliomyia hominivorax* (Coquerel) and *Lucilia sericata* (Meigen) *(Diptera: Calliphoridae)* [132,133,134,135,136,137], *Stomoxys calcitrans* (Linnaeus), *M. domestica*, *M. autumnalis* De Geer, *Haematobia irritans irritans* (Linnaeus) (Diptera: Muscidae) [138,139,140,141], *Glossina morsitans* Westwood, *G. pallidipes* Austen, *G. palpalis palpalis* (Robineau-Desvoidy), *G. tachinoides* Westwood (Diptera: Glossinidae) [142,143,144], *Boettcherisca peregrina* (Robineau-Desvoidy) (Diptera: Sarcophagidae) [145], *Zeugodacus cucurbitae* (Coquillett) and *C. capitata* (Diptera: Tephritidae) [146,147], and *Simulium* species (Diptera: Simuliidae) [148]. The AF of pteridines has been suggested to be a tool to assess their accumulation in the body of *Anopheles* mosquitoes, as supported by the results of extraction and High Performance Liquid Chromatography (HPLC) analysis procedures, by reading the emission at 455 nm under 365 nm excitation [149]. However, the quantification of AF levels in mosquitoes on the basis of spectrofluorometric approaches did not provide useful information on the age of the individuals, mostly likely because of too low pteridine levels and/or lack of methodological precision [150].

A note is also worth to be provided on the increase in the endogenous green AF detectable under blue light excitation in the eye and in the body of *Drosophila* soon after death [151]. The origin of this AF has yet to be clarified and it may be particularly useful to be further explored as potential biomarker of aging and death.

In general, being able to estimate insect age could be relevant since (i) it can help furnish a measure of the success of control measures, as a younger population age structure would indicate the desired increased adult mortality, (ii) in insect vectors, the probability to become infected increases with age [152,153], and (iii) it has a relevance for medical and forensic entomology, as it contributes to determine the postmortem interval [154,155].

### 2.4. High Resolution Morphology

Direct AF excitation, as well as the use of multi-photon AF excitation, second- and third-harmonic generation (SHG and THG, respectively) in the visible range and of near-infrared (NIR)-based approaches are particularly useful in achieving label-free imaging in samples with variable chemical composition. These techniques even allow deep tissue penetration (>500 µm), valuable for samples with nonlinear optical properties (see [156] for a review). Moreover, hyperspectral imaging (HSI), a technique able to acquire data simultaneously in hundreds of spectral bands with narrow bandwidths [157] has been recently applied also in entomological studies.

A label-free study performed on *Drosophila* larvae by using a 780 nm laser multiphoton excitation with the combined collection of both AF signal in the 435–700 nm ranges and SHG has provided a complementary structural information. Autofluorescence has evidenced internal organs such as the trachea and the digestive system, and SHG has revealed the larval muscles according to their repetitive structural organization, providing a promising supportive tool for combined physiological and developmental investigations [158].

Characterization of the larval morphology of various cyclorrhaphan species has been also achieved through confocal microscopy performed under couples of excitation/emission conditions at increasing wavelengths. The 3D organization of the structures composing the cephaloskeleton has been evidenced by the reconstruction of sequential, z stacking images collected under the different AF excitation/emission conditions (Figure 10) [159]. The use of both fresh or already prepared specimens from the National History Museum of London collection has indicated that KOH provided similar results, as compared with the clearing preparation involving the Hoyer’s medium containing chloral hydrate.

Sample clearing with KOH has been thus recommended as the preferred preparation procedure to obtain specimens suitable to be preserved for long time and thus for re-examinations by confocal microscopy, favoring sample exchange among scientists and comparative and taxonomic studies. The structural properties of insect cuticle represent indeed a strong limitation to internal tissues and organs imaging. In 2007, McGurk and colleagues described a method for 3D imaging of the intact adult *Drosophila* through its cuticle based on sample clearing, followed by optical projection tomography [160]. This approach allowed the observation of the gross anatomy of the fly based on AF, as a topographical reference in the localization of anatomical sites expressing GFP or a P-galactosidase fluorescing reporters following previous studies on the valuable use of AF to visualize *Drosophila* neuroanatomical structures in brain wax sections [161,162].

Ultramicroscopy reconstruction of *Drosophila* flight musculature, nervous system and digestive tract was then achieved [163], based on the set-up described by Becker and colleagues [164].

Subsequently, a double-side illumination strategy in ultramicroscopy has been developed, on the basis of the combination of aspheric lenses and optical elements to generate very thin light sheets. This approach applied to chemically-cleared *Drosophila* adult male individuals allowed to just use the AF of the sample to image both surface (e.g., hexagons of the lens facets, bristles on inter-ommatidial, maxillary palp and distal labellum) and internal (e.g., elements of the compound eye, the optic lobe, cardia) structures, to be made then observable with a highly sharp resolution in a 3D virtual reconstruction [165]. The 3D imaging analysis has been thus used as a technique additional to conventional optical or scanning electron microscopy, and its ability to provide variable views of the target organism has stimulated both new investigations to improve sample preparations, or applications to improve the detection of parasites (e.g., *Plasmodium* in infected *A. stephensi*) [36,166,167].

Multispectral imaging (MSI) and HSI have been successfully used in entomology to investigate insect pests and crop damages, examine stress responses, detect parasitoids and monitor the age of the sample with a daily precision, as for the case of hyperspectral measurements of the surface of the larvae of *L. sericata* [168]. Interestingly, HSI has been also used on trace immature blow flies (Diptera: Calliphoridae) development to determine their age for forensic applications [169]. Similarly, infrared spectroscopy based on attenuated total reflection-Fourier transform (ATR-FTIR) allowed to discriminate blow fly species at the larval stage [170].

In addition, the use of different models of convolutional neural networks to process RGB images collected by means of a digital camera from a light microscope was demonstrated able to classify, with different accuracy, the larvae of dipteran species belonging to the Calliphoridae and Muscidae families (Figure 11) [171]. The results of this study are highly promising for integration of data from artificial intelligence technology and biology to improve current approaches in forensic entomology.

Multispectral imaging has been also recently used to discriminate *A. fraterculus* (Wiedemann) (Diptera: Tephritidae) pupae for quality control purposes [172]. Interestingly, in this study MSI, which identifies the color differences in the pupae by capturing the visible region band, associated with NIR imaging able to distinguish the variations in the chemical composition of the samples, especially in terms of lipid content, allowed to detect and quantify pupal quality variations based on the reflectance patterns. This approach permitted to accurately classify the pupae in different quality-classes. Subsequently, the combination of hyperspectral and RGB imaging was shown to be able to differentiate between male and female *A. fraterculus* pupae, an issue with key importance for mass production and sterilization in Sterile Insect Technique (SIT) control programs [173]. Similarly, NIR imaging has been used to study intrapupal development in tsetse flies (Diptera: Glossinidae) and other Muscomorpha [174], with the potential to separate sexes days before emergence. Efficient sexing is highly desirable for operational SIT, as early sex sorting allows the males to be separately handled, irradiated and released [175,176]. Actually, NIR properties of Diptera are being increasingly explored in a range of applications. Indeed, NIR spectroscopy is capable of providing age-based classification also in mosquitoes [177,178,179,180,181,182,183,184]. Recently, NIR has been suggested as a useful tool to monitor the changes in age structure over time in an *Ae. aegypti* (Linnaeus) population, an aspect of key relevance for determining the effectiveness of vector control [185].

Near infrared spectroscopy has been also applied to evaluate the color variations and the patterns of microstructure in the scales of preserved specimens of *Sabethes* (*Sabethes*) *albiprivus* Theobald (Diptera: Culicidae) for species and subgenera classification (Figure 12) [186].

Moreover, mid-infrared (MIR) spectroscopy with a supervised machine learning has been used to accurately discriminate between vertebrate blood meals (e.g., human, chicken, goat or bovines) in the gut of *A. arabiensis* mosquitoes [187]. This application is particularly relevant for the rapid assessment of mosquito vectorial capacity and blood-feeding history.

Fluorescence imaging, in combination with composite imaging [188,189], can also be a powerful tool for comparative morphology of dipteran fossils preserved in different types of matrices due to the AF properties of the specimens. For example, a specimen of a trichocerid fly (Diptera, Hexapoda) from the Cretaceous Daya Formation in Kazakhistan has been found to be preserved in a matrix fluorescing under green light excitation (546 nm), making possible a high-quality imaging of the margins and surface of the wing in the completely black fossil [190]. More recently, larval and pupal fossil specimens preserved in Eocene Bitterfeld amber and Baltic ambers belonging to the dipteran ingroup Bibionomorpha were observed at 532 nm emission and described, taking advantage of the AF features of the samples [191]. This work greatly helped in gathering novel insights about the abundance of these larvae in their environment as well as on the related ecological data, contributing to further characterize the paleoecosystems preserved in amber matrices. Additional insights can be expected to improve the assessment of morphological details useful for taxonomic purposes, including potential reclassifications, as well as to facilitate exchange of both AF-based images and real specimens between laboratories and museums.

## 3. Conclusions

The current knowledge on the various fluorescing endogenous biomolecules that can be detected in Diptera, and on the various optical phenomena influencing the resemblance and visual sensing, derives from a huge amount of work, started several decades ago. In this review, we discuss the studies focused on the characterization of the most significant fluorescing biomolecules so far identified in this taxon (summarized in Table 1).

We also provide a summary of the technological applications of AF-based phenomena in Diptera, ranging from nanotube technology, in-flight detection of vectors and pests, non-invasive assessment of vectorial capacity, quality control in mass-rearing facilities, improved medical and forensic entomology, high-resolution morphology for comparative and taxonomic studies. Such comprehensive overview underlines a multifaceted field, which is still widely unexplored and holding great potential for both basic and applied science.

Not surprisingly, the majority of the studies we described have as target species *D. melanogaster*. This is of relevance, since *Drosophila* under living conditions and at different developmental stages is increasingly proving a promising tool not only to achieve a deeper understanding of development and metabolism in insects, but also to further exploit it as a model to study metabolic pathways and diseases of interest for human health. This would overcome the ethical issues growingly moved to the use of higher animals and tissue-derived models, such as organoids [208,209,210,211]. In addition, a continuously rising number of studies is focusing on exploring AF-based phenomena in mosquitoes and necrophagous flies, with important implications for medical and forensic entomology.

## Figures and Tables

**Figure 1 molecules-27-04458-f001:**
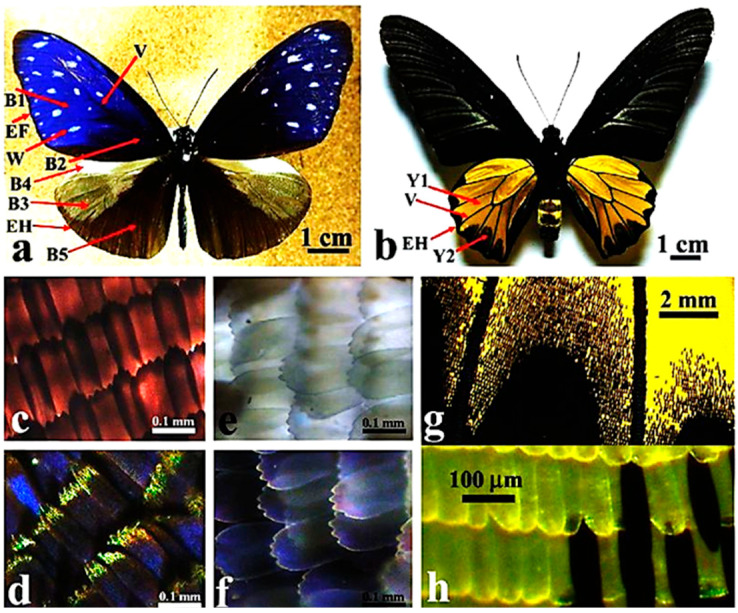
Dorsal side view of butterfly males of (**a**) *Euploea mulciber* Cramer and (**b**) *Troides aeacus* (Felder & Felder). Images of wing scales of *E. mulciber* observed at higher magnification under transmitted (**c**,**e**) and reflected (**d**,**f**) light conditions, from the blue (**c**,**d**) and white (**e**,**f**) regions indicated by B1 and W, respectively, in (**a**). Images of wing scales (indicated by Y2) of *T. aeacus* observed at variable magnification under transmitted (**g**,**h**) light conditions. The changes in color observed in *E. mulciber* in response to different illumination conditions have been explained with multiple interference effects induced by the layered arrangement of the brown scale cuticle, acting similarly to an optical diffraction grating. In *T. aeacus*, the yellow color does not show changes under different illumination conditions, consistently with the absence of multilayered rib-like wing scales and the consequent inability to produce scattering effects. Yellow is thus a real intrinsic color due to the presence of the pigment papilliochrome. Bars: 1 mm (**a,b**), 100 μm (**c,e,d,f**), 2 mm (**g**). Modified from [10].

**Figure 2 molecules-27-04458-f002:**
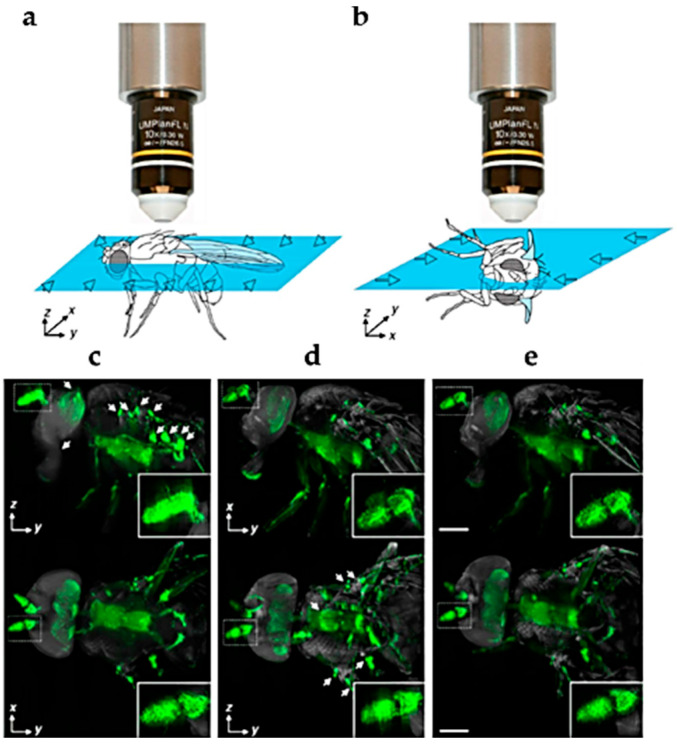
Representation of the orthogonal directions of light-sheet illumination to collect fluorescence signals through the *Drosophila* adult body (**a**,**b**), resulting in lateral ((**c**–**e**) upper panel) and dorsal ((**c**–**e**) lower panel) views. The multi-view reconstruction of combined stacks from (**c**,**d**) are shown in (**e**). Insets show the antenna at higher magnification. The topological localization of the GFP signal visible in green is facilitated by AF, representing in grey the overall body structures. Modified from [30]. Bars = 200 µm.

**Figure 3 molecules-27-04458-f003:**
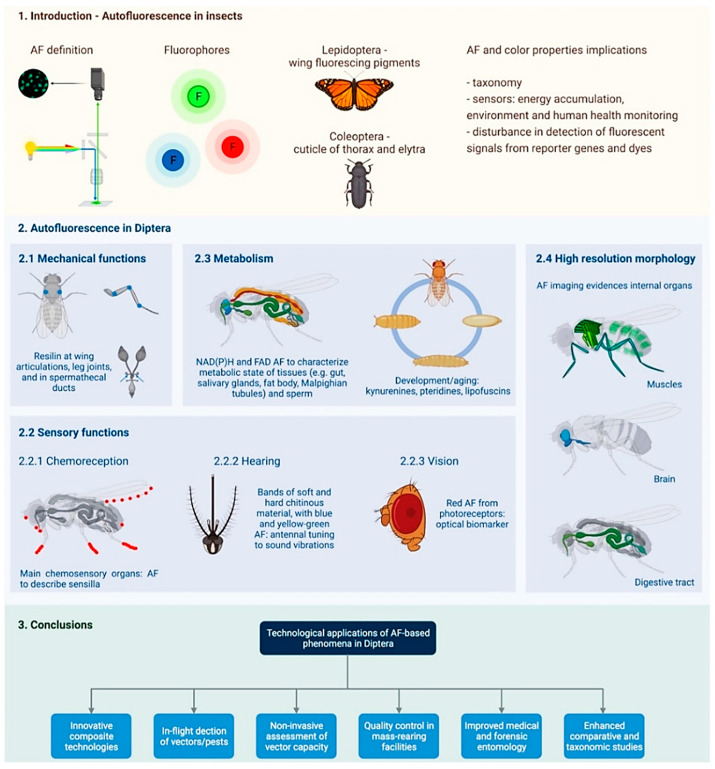
Schematic summary of the contents of this review. AF, autofluorescence. Created with BioRender.com (Accessed on 10 June 2022).

**Figure 4 molecules-27-04458-f004:**
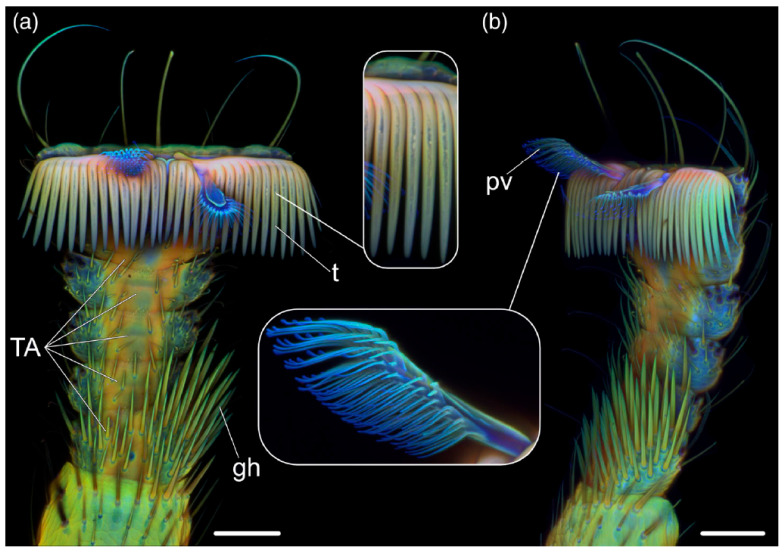
Images collected with a confocal laser scanning microscope from the leg at metathoracic position of *B. coeca*, under different AF measurement conditions favoring the bluish AF of resilin, or the yellowish or reddish AF of material at growing degrees of sclerotization. On these bases, the resilin-enriched pulvilli can be easily distinguished in the tarsomer (TA). (**a**) Ventral and (**b**) ventrolateral views; gh, grooming hairs; pv, pulvillus; t, claw tooth. Bar = 50 μm. Modified from [47].

**Figure 5 molecules-27-04458-f005:**
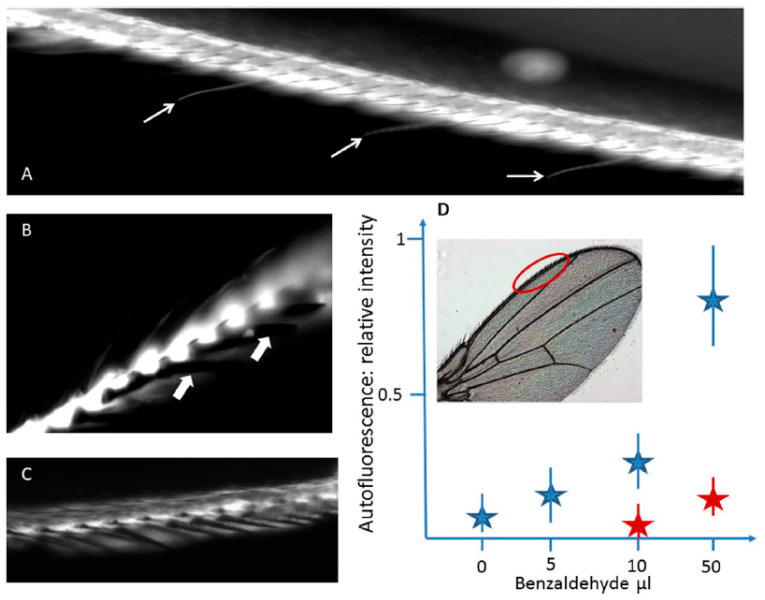
Treatment with volatile benzaldehyde results in the appearance of strong fluorescence emission in the regions of the wing of *Drosophila* corresponding to (**A**) the anterior margin of the vein, where also chemosensory sensilla are labelled (arrows), and (**B**) the chemosensory hairs, contrarily to the mechanoreceptors bristles (arrows). (**C**) In a cut wing subjected to the same treatment, the margin vein is labelled, contrarily to the thin chemosensory hairs. (**D**) Fluorescence intensity response in flies after a two-hour exposure to benzaldehyde at increasing concentrations, in live samples (blue stars; n = 10) or on cut wings (n = 5) treated under the same conditions (red stars). Modified from [67].

**Figure 6 molecules-27-04458-f006:**
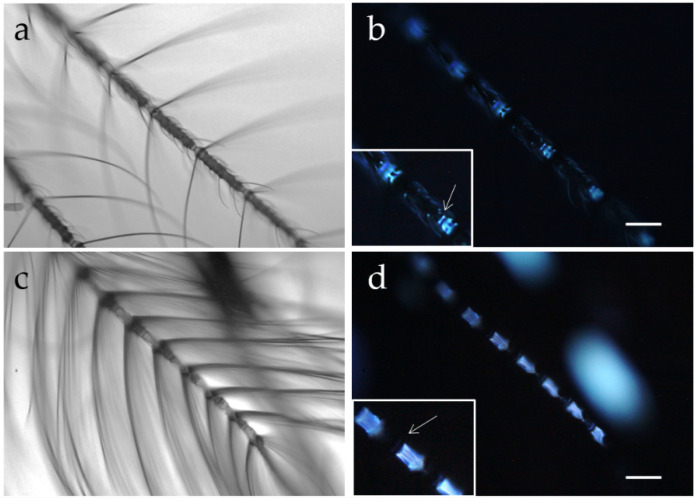
Antennae of female (**a**,**b**) and male (**c**,**d**) *Ae. albopictus* adults, observed under bright field (**a**,**c**) and fluorescence (**b**,**d**) microscopic conditions. More numerous and longer fibrillae are observed in males as compared with females, which have more sensilla distributed along the entire length of the antenna. Autofluorescence reveals an additional sexual dimorphism, showing a thin disc (arrow) near a larger cone (magnified in the inset) in each joint of the female antennae, while in the male a very thin blue disc (arrow) can be appreciated near the strong emission arising from the segments of the flagellomers 1–11 and showing a fibrous structural organization. These bluish emissions are relatable to the presence of resilin, as suggested by AF spectral analyses. Bars = 100 μm.

**Figure 7 molecules-27-04458-f007:**
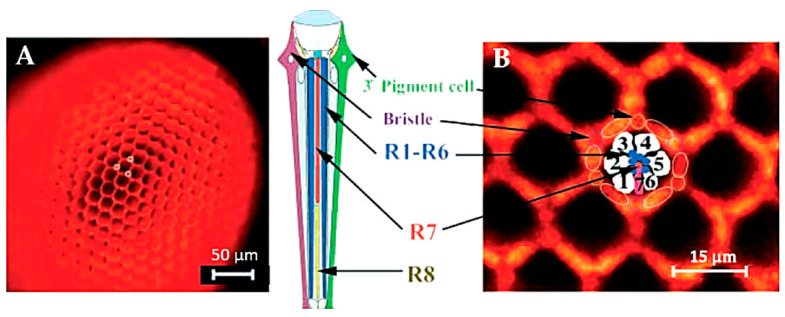
(**A**) Ommatidia of *Drosophila* eye imaged by epi-fluorescence under illumination with the green light. Under these observation conditions, the ordered honeycomb-like structure of the eye can be easily appreciated along with the accessory pigment cells. (**B**) Higher magnification shows the different cells composing an ommatidium pigment cells and lens facet, interpreted according to [92]. R1–R8, photoreceptor neurons. Modified from [91].

**Figure 8 molecules-27-04458-f008:**
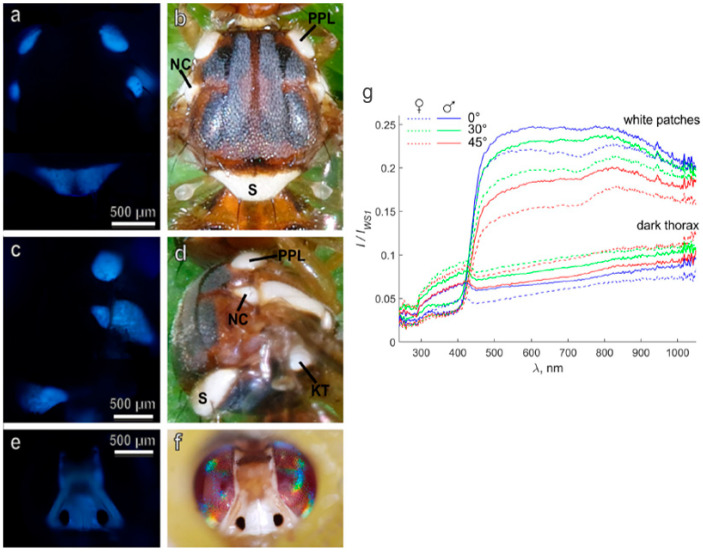
Images from the thorax (dorsal (**a**,**b**); lateral (**c**,**d**)) and the head (**e**,**f**) of *B. oleae*, captured under fluorescence (excitation 365 nm, emission > 397 nm; (**a**,**c**,**e**)) and bright field (**b**,**d**,**f**) light conditions. Notably, the bluish AF indicating the presence of resilin rises from the regions matching with the white patches; S, scutellum; PPL, post pronotal lobe; NC, notopleural callus; KT, katatergite. (**g**) Reflectance spectra recorded from dark areas and white patches of the surface of the thorax of *B. oleae* male and female individuals. Illumination was at 45°, and observed at variable degrees, identified by inset colors. Light scattering is due to the presence of biomaterial with several disordered surfaces and air-containing empty spaces in a phase of solid material, in this case represented by resilin. Modified from [97].

**Figure 9 molecules-27-04458-f009:**
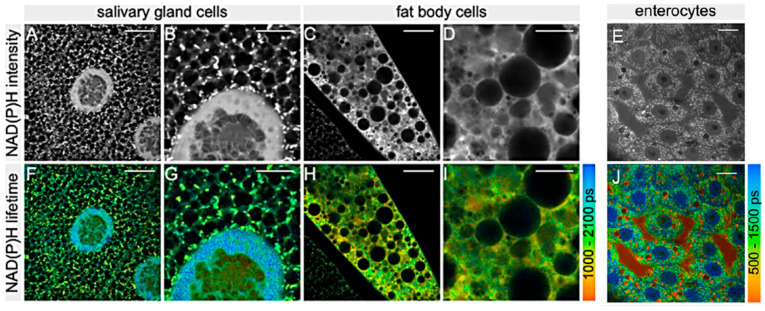
Fluorescence lifetime images recorded from salivary glands (**A**,**B**,**F**,**G**), fat body (**C**,**D**,**H**,**I**) and enterocytes (**E**,**J**) of *D. melanogaster*. Gray pictures represent the distribution intensity of the AF of NAD(P)H, and pseudo-color pictures represent the distribution of the mean lifetime values of NAD(P)H AF according to the respective color bars on the right; ps, picosecond. Bars = 10 μm (**A**,**F**,**C**,**H**,**E**,**J**), 5 μm (**B**,**G**,**D**,**I**). Modified from [102], with permission.

**Figure 10 molecules-27-04458-f010:**
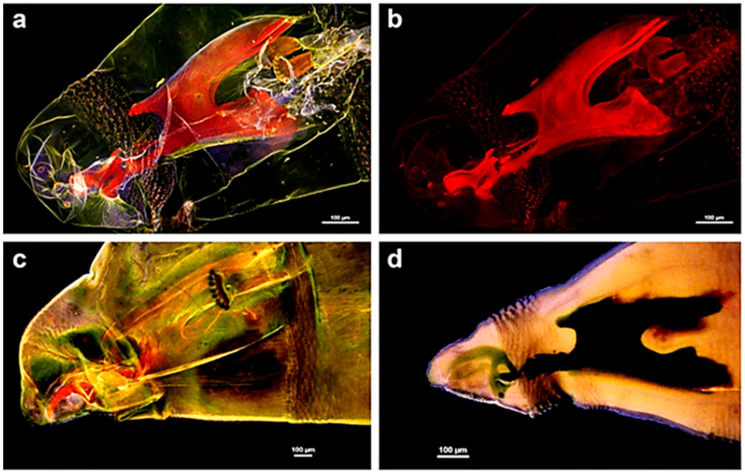
Cephaloskeleton of *L. sericata* (**a**–**c**) and *Muscina prolapsa* (Harris) (**d**) larvae. Optical sections recorded from *L. sericata* after clearing with KOH and Euparal embedding: (**a**) 27 optical sections from four lasers; (**b**) 27 sections from laser tuned at 640 nm; (**c**) 12 sections from four lasers. (**d**) *M. prolapsa* after Hoyer’s medium clearing, 11 optical sections from four lasers. The images show the importance of the proper choice of clearing procedures. Contrarily to KOH, Hoyer’s medium is unsuitable for confocal laser studies of AF since it prevents appreciation of cephaloskeleton AF, also because of emitted light absorption by soft tissues. Bars = 100 μm. Modified from [159].

**Figure 11 molecules-27-04458-f011:**
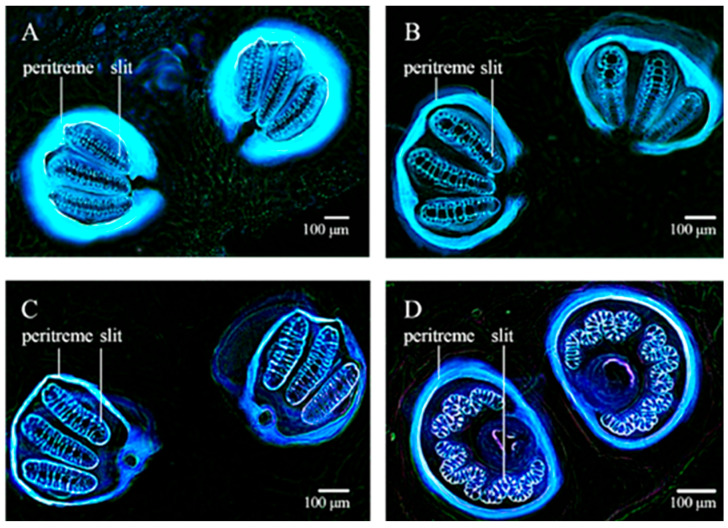
Inverted image colors showing the larval posterior spiracles of (**A**) *Chrysomya ruffifacies* (Macquart); (**B**) *C. megacephala*; (**C**) *L. cuprina*; (**D**) *M. domestica*. The morphological differences in the posterior spiracles confirm the important role of their characteristic structural organization as a tool for the identification of larvae of different dipteran species. Bars: 100 μm. Modified from [171].

**Figure 12 molecules-27-04458-f012:**
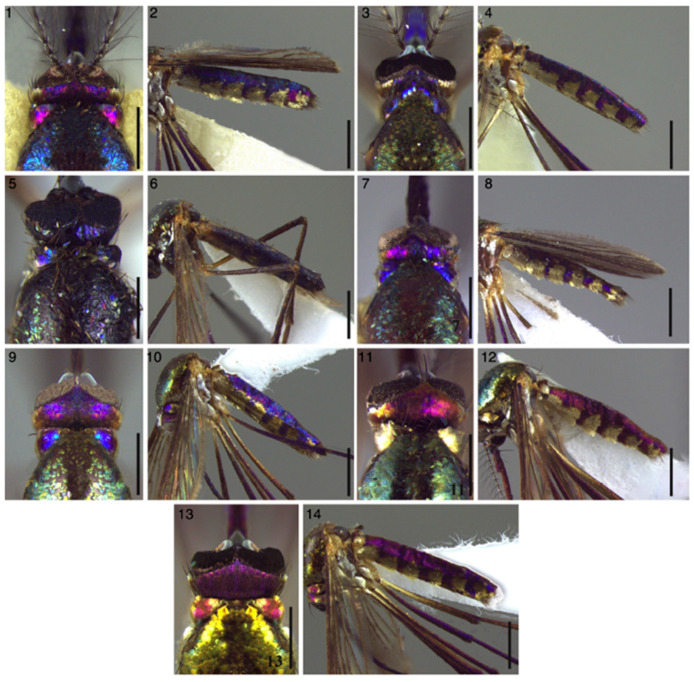
Examples of changes in color patterns characterizing the surface of different *Sa. albiprivus* specimens, which result from different types of reflection, i.e., purple, blue and green, or silver and gold, respectively. Bars = 1 mm (dorsal views, in images **1**, **3**, **4**, **5**, **7**, **9**, **11**, **13**), 0.5 mm (abdomen view, in images **2**, **6**, **8**, **10**, **12**, **14**). Color variations have been mostly ascribed to scale structural differences, assessed by scanning electron microscopy, and material composition, according to the results of principal component analysis performed by NIR analysis. Modified from [186].

**Table 1 molecules-27-04458-t001:** Fluorescing compounds in Diptera.

Biomolecules	Fluorophores	Functions	Absorption/ Excitation (nm)	Emission (nm)	Detection Methods	References
Resilin	Di-/tri-tyrosine residues, crosslinks and bridges	Structural protein/Biomechanical	320–380 (violet-blue)	400–500 (blue-green)	Spectroscopy/Wide-field, confocal microscopy imaging	[21,22,49,192,193]
Chitin and chitinous-mixed compounds	Undefined components	Polysaccharide semicrystalline biocomposite/Biomechanical	450–560 (blue-green)	>510–640 (green-red)	Spectroscopy/Wide-field, confocal microscopy imaging	[21,22,25,26,194]
	Optically active thin films		Scatter/Iridescence		Dual wavelength polarimetric spectroscopy/Reflectance, interference imaging	[195,196]
Rhodopsins	Metarhodopsin	Activated form of G-protein coupled receptor rhodopsin/Vision	300–400; 400–650 (violet); (blue-red)	600–750 (red-deep red)	Spectroscopy/Wide-field microscopy imaging	[90,91,93,197,198]
Pyridinic derivatives	NAD(P)H _bound/free_	Coenzymes in redox reactions in energy metabolism, reductive biosynthesis	330–380 (violet)	440_bound_; 480_free_ (violet-blue); (blue)	Spectroscopy/Wide-field microscopy imaging, multiphoton lifetime microscopy imaging	[101,102,107,108,109,199,200]
Riboflavin derivative	FAD		360/445 (violet)/(blue)	480/540 (blue)/(green)		
Intermediate metabolites of tryptophan	Kynurenine and derivatives	Nitrogen excretion pathway; light screening in eyes; aging markers	300–390/380–420 (violet)/(violet-blue)	430–550 (blue-yellow)	Spectroscopy/Wide-field microscopy imaging	[113,114,115,116,128,131,201,202,203,204]
Pterine derivatives	Pteridines		355 (violet)	400–550 (blue-yellow)		
Peroxidized lipids, proteins, carotenoids and their mixed compounds	Lipofuscin-like lipopigments	Aging and oxidative stress markers	Near UV—~500 (violet-green)	>460 (blue-red)	Spectroscopy/Wide-field microscopy imaging	[118,120,124,125,205,206,207]

The wavelength range for excitation/emission may change in dependence on the microenvironment and/or the biochemical complexity of some of the reported biomolecules, and on the detection conditions applied. Representative spectra of the fluorophores are given in cited literature.

## Data Availability

Not applicable.

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
