# Peer review of "Autofluorescent Biomolecules in Diptera: From Structure to Metabolism and Behavior"

_molecules, 2022, doi:10.3390/molecules27144458_

Round 1
Reviewer 1 Report
Very interesting review on the autofluorescence pigments in flies/ Diptera! Overall it is nicely structured but I would recommend making some revisions:
1. Is it possible to have a table of contents at the beginning of the review? This would help a lot. For instance, I looked for autofluorescence in the gut but didn't find special section...
2. For the table 1, I would suggest to make some revision. Perhaps it is worth adding the spectrum with the dyes (or placing them from 'blue' to 'red') and listing the measurement modalities? I see that very different imaging approaches are used and it's a bit confusing which pigment can be detected by which method.
3. If possible, add minor description of microscopy/ macroscopy or related methods, which are used to detect these pigments and what can be achieved as a readout. For instance, fluorescence lifetime imaging microscopy is used for metabolism (NADH/FAD) but the method is not well described as e.g. in Dmitriev, Intes, Barroso, J Cell Sci 2021 or Torrado, Malacrida and Ranjit, Sensors, 2022 or Alfonso-Garcia, Bec, Weyers, Marsden, Zhou, Marcu, J Biophotonics 2021.
4. Is there additional application area for FLIM in Diptera, apart from NADH/FAD (metabolism)? I wonder if the gut autofluorescence or from other pigments can be also used.
5. Does the autofluorescence of fossil diptera samples also have practical applications? Would be interesting to see some thoughts on it.
Author Response
Responses to Reviewer 1 Comments
We have greatly appreciated the positive comments of Reviewer # 1 about our work, and we thank him/her very much for the time spent working on our text and for providing constructive remarks, which surely helped us to improve the paper.
Q1. Is it possible to have a table of contents at the beginning of the review? This would help a lot. For instance, I looked for autofluorescence in the gut but didn't find special section.
A1. A new Figure (Figure 3) has been added to the review, to provide a schematic summary of the content of the review and facilitate searching of specific aspects. Subsequent Figures have been renumbered accordingly.
Q2. For the table 1, I would suggest to make some revision. Perhaps it is worth adding the spectrum with the dyes (or placing them from 'blue' to 'red') and listing the measurement modalities? I see that very different imaging approaches are used and it's a bit confusing which pigment can be detected by which method.
A2. The Table 1 has been duly revised to address the Reviewer’s comments, incorporating also the Reviewer’s 2 requests, and has been now moved to the end of the text. The columns for biomolecules / fluorophores / functions have been reorganized. For each pigment the colors for the spectral ranges for the proper excitation and detection of the emission have been added to the column already reporting the wavelengths, and the detection methods have been listed. In principle, the detection of each pigment depends on the use of the proper light conditions for excitation and detection of the emission. The new organization of the Table should now clearly represent that different measurement approaches, i.e. spectroscopy and various imaging modalities, can allow the detection of a pigment by using the proper light illumination and reading.
Q3. If possible, add minor description of microscopy/ macroscopy or related methods, which are used to detect these pigments and what can be achieved as a readout. For instance, fluorescence lifetime imaging microscopy is used for metabolism (NADH/FAD) but the method is not well described as e.g. in Dmitriev, Intes, Barroso, J Cell Sci 2021 or Torrado, Malacrida and Ranjit, Sensors, 2022 or Alfonso-Garcia, Bec, Weyers, Marsden, Zhou, Marcu, J Biophotonics 2021.
A3. New text and related references have been now added to the manuscript, to briefly describe microscopy/ macroscopy or related methods (lines 40-61; 100-121; 508-528, please refer to tracked change version).
Q4. Is there additional application area for FLIM in Diptera, apart from NADH/FAD (metabolism)? I wonder if the gut autofluorescence or from other pigments can be also used.
A4. To our knowledge, these is no other application areas of FLIM in Diptera than NAD(P)H/FAD for energy metabolism, as already given in the text. Only a report is given about resilin and chitin from Reinhardt et al., 2017, but on the bedbug Cimex lectularius (lines 616-618, please refer to tracked
change version). As to autofluorescence of the gut, besides the FLIM based study on NAD(P)H in Drosophila enterocytes (Ref. 102, Wetzker C and colleagues, 2019), the ancient report on the autofluorescence of lipofuscins detected in the oenocytes of the midgut adult male housefly has been added (Ref. 125, Donato H& Sohal RS, 1978).
Q5. Does the autofluorescence of fossil diptera samples also have practical applications? Would be interesting to see some thoughts on it.
A5. The reviewer’s comment has been addressed and the text has been expanded accordingly (lines 794-809, please refer to tracked change version).

Reviewer 2 Report
The manuscript by Anna C. Croce and Francesca Scolari reviews an issue of autofluorescence characteristic of the tissues of insects from the order Diptera. Within the first chapters, the authors make an excursus into the molecular bases of insect pigmentation and the optical properties of their structural components, moving on to a narrower topic—endogenously fluorescent molecules in Diptera. Next, there is a comprehensive analysis of the entomological challenges (including studies of the anatomy, morphology, physiology, and biochemistry of dipteran species) addressed by the autofluorescence imaging of insects’ tissues. It should be noted that the authors provide rich details on the specific imaging techniques used in the relevant studies, thus making the review’s material interdisciplinary and useful both for entomologists/insect biochemists and for researchers in the field of fluorescence imaging. The manuscript can certainly be recommended for publication in nearly ‘as-it-is’ form. I have found just a few minor issues:
1) To my mind, column ‘Biochemical features' in table 1 (page 5) would benefit from some extension with either schematics or description of the chemical structure of chromophores involved in AF. While the nature of light-absorbing group is clear in the case of flavins or opsins, that of resilin protein (is it a tryptophan fluorescence?) or lipofuscins may raise questions. It is also a bit confusing that in column 'Functions’, one finds a mixture of the biological functions and scientific applications. The authors could consider the column’s renaming;
2) Italics use for the specific names is not standardized throughout the text;
3) line 430 ‘Lifetime imaging base on the selective detection…’ probably it should be ‘based’;
4) caption of Fig.8, lines 441-443 ‘Gray pictures represent the distribution intensity of NAD(P)H, and pseudo-color pictures represent the distribution of the mean lifetime values of NAD(P)H’. Slightly incorrect phrasing. Intensity and mean lifetime characterize fluorescence of NAD(P)H, not NAD(P)H as is;
5) line 652 ‘Indeeed’ instead of ‘indeed’.
Author Response
Responses to Reviewer 2 Comments
We have greatly appreciated the positive comments of Reviewer # 2 about our work, and we thank him/her very much for the time spent working on our text and for providing constructive remarks, which surely helped us to improve the paper.
Q1. To my mind, column ‘Biochemical features' in table 1 (page 5) would benefit from some extension with either schematics or description of the chemical structure of chromophores involved in AF. While the nature of light-absorbing group is clear in the case of flavins or opsins, that of resilin protein (is it a tryptophan fluorescence?) or lipofuscins may raise questions. It is also a bit confusing that in column 'Functions’, one finds a mixture of the biological functions and scientific applications. The authors could consider the column’s renaming;
A1. The Table 1 has been duly revised, to address the Reviewer’s comments, incorporating also the requests from Reviewer 1.
Q2. Italics use for the specific names is not standardized throughout the text;
A2. We used the following common rule for species abbreviations: genus name written in full upon its first use; for subsequent uses, we abbreviated the genus to its first letter also when describing a different species within that genus, since there was no risk of confusing it for another genus. We also modified the text to report the complete species name inclusive of the scientist who first described it. We also went through the text and fixed any italics issues.
Q3. line 430 ‘Lifetime imaging base on the selective detection…’ probably it should be ‘based’;
A3. Corrected.
Q4. caption of Fig.8, lines 441-443 ‘Gray pictures represent the distribution intensity of NAD(P)H, and pseudo-color pictures represent the distribution of the mean lifetime values of NAD(P)H’. Slightly incorrect phrasing. Intensity and mean lifetime characterize fluorescence of NAD(P)H, not NAD(P)H as is
Q4. Corrected.
Q5. line 652 ‘Indeeed’ instead of ‘indeed’.
A5. Corrected.
